# Improved Fine-Tuning by Better Leveraging Pre-Training Data

**Ziquan Liu**[1,*] **Yi Xu**[2✉], **Yuanhong Xu**[3], **Qi Qian**[3], **Hao Li**[3], **Xiangyang Ji**[4], **Antoni B. Chan**[1], **Rong Jin**[3]

[1]Department of Computer Science, City University of Hong Kong
[2]School of Artificial Intelligence, Dalian University of Technology
[3]DAMO Academy, Alibaba Group
[4]Department of Automation, Tsinghua University
`ziquanliu2-c@my.cityu.edu.hk`, `yxu@dlut.edu.cn`, {`yuanhong.xuyh`, `qi.qian`,
`lihao.lh`}`@alibaba-inc.com`, `xyji@tsinghua.edu.cn`, `abchan@cityu.edu.hk`,
`rongjinemail@gmail.com`

## Abstract

As a dominant paradigm, fine-tuning a pre-trained model on the target data is widely used in many deep learning applications, especially for small data sets. However, recent studies have empirically shown that training from scratch has the final performance that is no worse than this pre-training strategy once the number of training samples is increased in some vision tasks. In this work, we revisit this phenomenon from the perspective of generalization analysis by using excess risk bound which is popular in learning theory. The result reveals that the excess risk bound may have a weak dependency on the pre-trained model. The observation inspires us to leverage pre-training data for fine-tuning, since this data is also available for fine-tuning. The generalization result of using pre-training data shows that the excess risk bound on a target task can be improved when the appropriate pre-training data is included in fine-tuning. With the theoretical motivation, we propose a novel selection strategy to select a subset from pre-training data to help improve the generalization on the target task. Extensive experimental results for image classification tasks on 8 benchmark data sets verify the effectiveness of the proposed data selection based fine-tuning pipeline. Our code is available at https://github.com/ziquanliu/NeurIPS2022_UOT_fine_tuning.

## 1 Introduction

After the success on ImageNet [1], deep learning attracts much attention and improves the performance of various computer vision tasks significantly, e.g., object detection [2], semantic segmentation[3]. However, as a result of expensive labeling, it is unlikely that we have sufficient labels for every application. Fortunately, given a model pre-trained on a large-scale data set like ImageNet, an effective model for the target data set, which may only have hundreds of examples, can be learned by fine-tuning the pre-trained model. It is because many vision tasks are related [4] and a model learned from ImageNet that consists of more than one million examples can contain diverse semantic information and provides a better initialization than random initialization.

Despite the prevalence of fine-tuning pre-trained models, its theoretical understanding is unclear. On the one hand, when sufficient training data is available, some research [5] has shown that training

---

*Work partially done during an internship at DAMO Academy, Alibaba Group. ✉ indicates corresponding author.

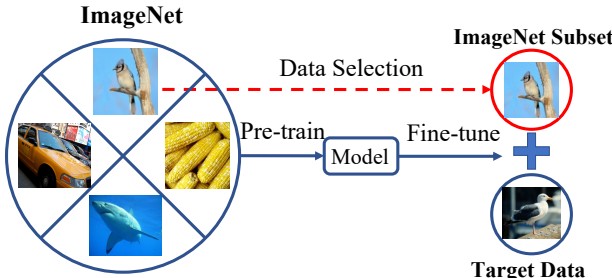

Figure 1: The proposed pre-training data reusing method, which is motivated by our generalization analysis of the effect of pre-training data on fine-tuning. A novel data selection method based on unbalanced optimal transport is proposed to reuse the appropriate pre-training data in fine-tuning. Averaged over 8 classification data sets, the UOT-selection method improves the performance of vanilla fine-tuning with a large margin of 2.93% using a self-supervised pre-trained model.

from scratch can achieve the same accuracy as initialization with ImageNet pre-trained models after additional training. As a warm-up of this work, we aim to understand this phenomenon from the theoretical side of generalization analysis [6]. Our analysis reveals that the final prediction precision may have a weak dependency on the pre-trained model when the target training set is large enough. On the other hand, it is not realistic to have abundant labeled data for every target task. So in many vision applications, training from scratch generally cannot match the performance of fine-tuning a pre-trained model. However, our theoretical result tells us that when the pre-training data are too far from the target data, the domain gap will hurt the accuracy of target tasks.

These two observations lead to the following question: can we develop a new strategy of *fine-tuning* that achieves better generalization performance than the standard framework by reusing pre-training data in fine-tuning to reduce the domain gap? This work will address this question with an affirmative answer. Inspired by the theoretical observation, we propose to leverage the pre-training data, which is also available for fine-tuning, for target tasks. Concretely, we propose to reuse pre-training data and optimize its task loss (e.g., cross-entropy loss for classification task) along with the target data when fine-tuning. The generalization analysis confirms that the performance on the target data can be improved when an appropriate portion of pre-training data is selected, as illustrated in Figure 1.

Since target data can be from different domains, we study the reusing strategy of pre-training data for different cases. First, when the target data is closely related to the pre-training data, one can randomly sample a number of pre-training data for fine-tuning, which is referred to as **random selection**. Second, if the label information of pre-training data is available and the classes overlapped with target data are identifiable, one can directly use those data with overlapped classes in fine-tuning. For example, given the data set of CUB [7], which consists of birds images, 59 bird classes [8] in ImageNet can be reused in fine-tuning. This scheme is referred to as **label-based selection**. Finally, when the labels between pre-training and target domains cannot match exactly or pre-training data has no labels as in self-supervised pre-training, the similarity measured by representations extracted from the corresponding pre-trained model will be adopted for selection. The last setting is prevalent in real-world applications and referred to as **similarity-based selection**.

Given the large scale of pre-training data, the representations from the pre-trained model can capture semantic similarity [9]. Based on this observation, we propose a novel selection algorithm to obtain a subset from pre-training data closest to the target data by solving an unbalanced optimal transport (UOT) problem. Interestingly, the proposed method performs consistently well on other scenarios, e.g., labels are overlapped, which reduces the effort of identifying overlapped pre-training classes. The main contributions of this work are summarized as follows.

- From the perspective of generalization analysis, this work explains the phenomenon that training from scratch has a similar final performance as fine-tuning the pre-trained model in some computer vision tasks, when the training data in the target domain is sufficient.
- We develop the generalization analysis when pre-training and target data are used in fine-tuning simultaneously, under some mild assumptions. It demonstrates that the performance on target data will likely be improved when the pre-training data is similar to the target data.
- With the insight of generalization analysis, we propose to select a subset of pre-training data with better similarity to the target data to further boost the final performance. A novel UOT-based algorithm is developed to handle target data from different scenarios.

- The performance of the proposed fine-tuning process is evaluated on 8 benchmark data sets for image classification tasks. When a self-supervised pre-trained model is used, our method, UOT fine-tuning, surpasses the conventional fine-tuning pipeline by a large margin of 2.93% averaged over all tasks, verifying the effectiveness of reusing pre-training data.

## 2 Related Work

Fine-tuning as a special case of transfer learning [10, 11, 12, 13] aims to improve the performance on the target data by transferring the knowledge from a large-scale pre-training data to a target domain. For example, supervised pre-trained models on ImageNet have been extensively used in image classification [9], object detection [2, 14] and semantic segmentation [3, 15]. However, the empirical study in [5] shows that the advantage of a supervised pre-trained model over random initialization cannot be observed when the gap between pre-training and target task is large, or the target task has sufficient training data and is trained for sufficient time. Later, [4] demonstrates that self-supervised pre-training improves upon training from scratch in object detection and other vision tasks with strong data augmentation, indicating that self-supervised pre-training learns more general visual representations. Our work considers a general pre-training paradigm including both supervised and self-supervised approaches, and explains why pre-trained models fail to significantly outperform random initialization from the view of generalization theory. Different from existing work that regularizes the fine-tuning optimization explicitly [16, 17], we propose to reuse pre-training data in target training based on the theoretical findings.

There are several existing papers that explore the source data selection [18, 19, 20]. [18] improves fine-tuning by borrowing data from a source domain which is similar to the target domain. The difference between [18] and our work are two-fold: 1) our work proposes a novel pre-training and fine-tuning pipeline while [18] is a joint training framework *without pre-training*; 2) our proposed data selection is a global search method using deep features from the pre-training model while [18] uses low-level features and retrieves similar images with local search. Our paper demonstrates the weakness of local search compared to global search in our experiment. [19, 20] proposed similar schemes that pre-train the model on the selected subset from the pre-training data according to a domain similarity measure, but they do not use the selected data along with target data in the fine-tuning but *re-pre-train* on the selected data for every target task, which is a fundamental difference between their works and ours. It is not surprising that such a re-pre-training framework brings benefit to a target task but it costs more computational time and resources than our *fine-tuning* framework. Another type of works uses the relationship between source and target to improve fine-tuning [21, 22]. [21] exploits the relationship between source and target labels, and [22] trains a policy network to control the gradient mask for backbone's blocks. Our paper explicitly adds a selected portion of pre-training data in fine-tuning and the selection strategy can handle self-supervised pre-training since it does not need source labels. [23] proposes to optimize a weight for the target data samples instead of selecting pre-training data for improving fine-tuning. Such target-weighting methods are compatible with ours, which selects source data, and future work will consider their combination.

The similarity-based data selection scheme in this work is the one based on a variant of optimal transport (OT) optimization. General OT is often used in computer vision to estimate or/and minimize the distance between two probability measures, such as prediction probabilities in classification [24], density maps in crowd counting [25] and the reconstruction loss in generative models [26, 27]. [28] measures the distance between two data sets by using OT and label information. Our paper solves an unbalanced optimal transport (UOT) problem between pre-training and target data to obtain a similarity vector for pre-training data to select a portion of data close to the target task.

## 3 Main Results

### 3.1 Theoretical Understanding

**Preliminary.** The target problem of interest that we aim to optimize can be formulated as

$$\min_{\theta \in \mathbb{R}^d} F(\theta) := \mathrm{E}_{(x,y)\sim\mathbb{P}} \left[ f(\theta; x, y) \right], \tag{1}$$

where $\theta$ is the model parameter to be learned; $(x, y)$ is the input-label pair that follows a unknown distribution $\mathbb{P}$; $\mathrm{E}_{(x,y)\sim\mathbb{P}}[\cdot]$ is the expectation that takes over a random variable $(x, y)$ while we use

$E[\cdot]$ for the sake of simplicity when the randomness is obvious; $f(\cdot; x, y)$ is a loss function. Suppose we have a set of training data $\{(x_1, y_1), \ldots, (x_n, y_n)\}$ drawn from $\mathbb{P}$ are given, where $n$ is the sample size. In practice, we want to solve the following empirical version of problem (1):

$$\min_{\theta \in \mathbb{R}^d} F_n(\theta) := \frac{1}{n} \sum_{i=1}^n f(\theta; x_i, y_i). \tag{2}$$

Stochastic gradient descent (SGD) [29] is a very popular algorithm for solving problem (2) in many computer vision tasks, whose updating is given by $\theta_{t+1} = \theta_t - \eta \nabla_\theta f(\theta_t; x_{i_t}, y_{i_t})$, where $t = 0, 1, \ldots$, $\eta > 0$ is the learning rate, $\nabla_\theta f(\theta; x, y)$ is the gradient of function $f(\theta; x, y)$ with respect to variable $\theta$. When the variable to be taken a gradient is obvious, we use $\nabla f(\theta; x, y)$ for simplicity. We use the excess risk (ER) as the performance measurement for a solution $\widehat{\theta}$: $F(\widehat{\theta}) - F(\theta_*)$, where $\theta_* \in \arg\min_{\theta \in \mathbb{R}^d} F(\theta)$ is the optimal solution of (1) and $\widehat{\theta}$ is the output of SGD.

In order to describe the pre-trained model, we denote by $G(\theta) := \mathrm{E}_{(x', y') \sim \mathbb{Q}}[g(\theta; x', y')]$ the objective function that the pre-trained model aims to optimize. We also suppose that we have a set of training data $\{(x_1', y_1'), \ldots, (x_m', y_m')\}$ drawn from $\mathbb{Q}$. Usually, the sample size of pre-training data is larger than that of target data, i.e., $m \gg n$. For the sake of analysis, we let $m$ be large enough and both the pre-trained model and the target learning task share the same set of parameters. In order to ensure that the model learned by optimizing $G(\theta)$ will be valuable to the optimization of $F(\theta)$, we assume that the difference of their gradients is bounded. That is, there exists a constant $\Delta > 0$ such that $\|\nabla F(\theta) - \nabla G(\theta)\| \leq \Delta, \forall \theta \in \mathbb{R}^d$.

**Value of Pre-trained Model.** To see the value of a pre-trained model, we present the excess risk bounds of the pre-trained model $\theta_p$ and the final model $\theta_f$ after fine-tuning $\theta_p$ for the target task in the following lemma.

**Lemma 1** (Informal). *We have the following performance guarantees for target task $F(\theta)$ in expectation. (1) The pre-trained model $\theta_p$ provides $F(\theta_p) - F(\theta_*) \leq O(\Delta^2)$. (2) The final model $\theta_f$ after fine-tuning $\theta_p$ against a set of $n$ training examples provides $F(\theta_f) - F(\theta_*) \leq O\left(\frac{\log(n\Delta^2)}{n}\right)$.*

First, the performance gap between pre-trained model $\theta_p$ and the optimal model $\theta_*$ is bounded by $O(\Delta^2)$, where $\Delta$ describes the approximation accuracy when replacing $\nabla F(\theta)$ with $\nabla G(\theta)$. Second, note that $\Delta$ only appears in the logarithmic term, implying that the excess risk bound has a weak dependency on the pre-trained model when $n$ is large. That is to say, when $n$ is lager, the pre-trained model has a small effect on the final performance, which is consistent with the empirical results found in [5].

**Value of Pre-training Data.** In reality, a target task often does not have enough data to fine-tune the model for a long time, so the pre-trained models are still better than random initialization in many vision tasks. Nevertheless, Lemma 1 reveals that even $n$ is not large, the dependency of generalization on a pre-trained model is potentially weak. This leads to a natural question: is it possible to design a better fine-tuning process that can overcome the limitation of the existing one? To this end, we develop a new approach for fine-tuning that aims to leverage the data used for the pre-trained model during the phase of fine-tuning. In this way, we aim to solve the following problem during the fine-tuning process:

$$\alpha F_n(\theta) + (1 - \alpha) H_m(\theta), \tag{3}$$

where $\alpha \in (0, 1]$ is a constant, $F_n(\theta)$ is defined in (2), and $H_m(\theta) := \frac{1}{m} \sum_{j=1}^m h(\theta; x_j', y_j')$ where $h$ is a loss function related to target task and $\xi_j := (x_j', y_j')$ is drawn from $\mathbb{Q}$. Note that the loss function $h$ can be same as the loss function of the target task. The solution is updated by SGD: $\theta_{t+1} = \theta_t - \eta \nabla \widetilde{f}(\theta_t)$ where $\nabla \widetilde{f}(\theta_t)$ is the stochastic gradient related to (3). The theorem below provides a performance guarantee for fine-tuning via (3).

**Theorem 1** (Informal). *We have the following performance guarantee for target task $F(\theta)$ in expectation,*

$$F(\theta_{f_*}) - F(\theta_*) \leq O\left(\frac{\alpha \log(n\Delta^2/\alpha)}{n} + (1 - \alpha)\delta^2\right), \tag{4}$$

*where $\delta^2 := \max_{\theta_t, \xi_{i_t}}\{\mathrm{E}[\|\nabla F(\theta_t) - \nabla h(\theta_t; \xi_{i_t})\|^2]\}$ and $\theta_{f_*}$ is the final model for the target task.*

When $\delta^2$ is small, by choosing appropriate $\alpha \in (0, 1]$, we may be able to further reduce the error from $F(\theta_{f_*})$. When $\delta^2$ is large, that is, when the second term of bound in (4) dominates the total

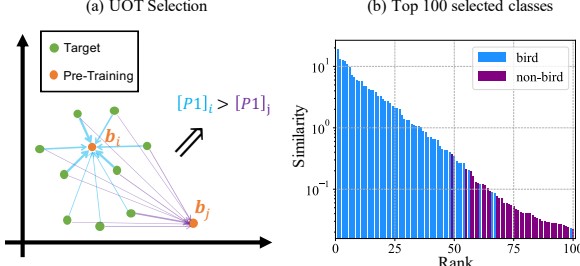

(a) UOT Selection          (b) Top 100 selected classes

Figure 2: Illustration of UOT selection. **(a)** UOT selection. Green dots denote target classes and orange dots denote pre-training classes/clusters. Blue and purple arrows show the result of UOT for $\mathbf{b}_i$ and $\mathbf{b}_j$, where the thickness means similarity. **(b)** Top-100 similarity in $\mathbf{P1}$ on CUB data set. Most bird classes from ImageNet are selected in the top-100 similarity vector to CUB.

error, then it would be worse than the result of standard fine-tuning a pre-trained model in Lemma 1. Therefore, our goal is to select appropriate $\xi_{i_t}$, the training examples from pre-training data, such that $\nabla h(\theta_t; \xi_{i_t})$ can better approximate $\nabla F(\theta_t)$. These theoretical observations inspire us to design a selection strategy for pre-training data, that is, to select images *similar* to those of target data from pre-training data and use these selected images during fine-tuning.

## 3.2 The Proposed Data Selection Strategy

Theorem 1 shows that the benefit of a pre-trained model can be enhanced when pre-training data are used during the fine-tuning process. This inspires us to propose data selection strategies and to choose an appropriate portion of pre-training data. In experiments, we follow the standard pre-training practice in computer vision to use a deep neural network pre-trained on ImageNet, and then select data from ImageNet to help fine-tuning on target classification tasks. We summarize the proposed pre-training data reuse strategies as follows.

**Label-based Selection**  When the label information of pre-training data is available, and the overlapped classes with target data are recognizable, one can simply select the overlapped classes and use them during the fine-tuning. For instance, the bird images from ImageNet are all selected when fine-tuning CUB. Unfortunately, this scheme heavily depends on the label match between pre-training and target data, which may worsen the performance in some real-world applications without perfectly matched classes.

**Random Selection**  The second data selection scheme is to choose classes with uniform sampling, referred to as random selection. This strategy can improve the performance of target tasks if the domain gap $\delta^2$ between pre-training and target data is small, and keep the weights close to initialization if the selected data are sufficiently large. The drawback of uniform selection is that the domain gap $\delta^2$ is not considered in the data reusing process, so the performance heavily depends on the inherent property of data sets.

**Similarity-based Selection**  To reduce the domain gap, we propose the third data selection scheme, an UOT-based method, to choose data classes from the pre-training set whose distributional distance to the target data set is small. The UOT-selection method is able to handle pre-training data with and without labels. When a supervised pre-trained model is used for fine-tuning, there are class labels for pre-training images so the selection unit of UOT is class. When a self-supervised pre-trained model is used and there are no labels for pre-training images, we index the pre-training data by clustering in the feature space of the corresponding pre-trained backbone. Therefore, for unlabeled pre-training data, the selection unit is cluster.

With the labels or cluster indices, each class/cluster is represented as the mean of deep features from the pre-trained model, e.g., 512-dim features from the penultimate layer of a pre-trained ResNet18 model. Since the training set often has balanced classes, all classes or clusters are assigned with unit weights for both pre-training and target set. So we have two density measures for the target set and pre-training set, i.e. $\{(\mathbf{a}_i, w_i^{(f)} = 1)\}_{i=1}^{K_f}$ and $\{(\mathbf{b}_j, w_j^{(g)} = 1)\}_{j=1}^{K_g}$, respectively. Denote the features of target and pre-training data as $\mathbf{v}_i^{(f)}$ and $\mathbf{v}_j^{(g)}$, $\mathbf{a}_i = \sum_{y_s=i} \mathbf{v}_s^{(f)}/n_i^{(f)}$ and $\mathbf{b}_j = \sum_{y_t=j} \mathbf{v}_t^{(g)}/m_j^{(g)}$, where $n_i^{(f)}$ is the number of images in $i$-th class of target data and $m_j^{(g)}$ is defined similarly for pre-training data. In the general case where $K_f \neq K_g$, the two measures have different total masses

so we propose to compute the unbalanced OT distance between the two by a generalized Sinkhorn iteration [30]. Specifically, the optimization objective is formulated as a UOT problem,

$$\min_{\mathbf{P} \in \mathbb{R}_+^{K_g \times K_f}} \langle \mathbf{P}, \mathbf{C} \rangle - \epsilon h(\mathbf{P}) + \tau_1 KL(\mathbf{P}\mathbf{1}, \mathbf{w}^{(g)}) + \tau_2 KL(\mathbf{P}^T\mathbf{1}, \mathbf{w}^{(f)}), \tag{5}$$

where $\mathbf{C}_{i,j}$ is the distance between $\mathbf{a}_i$ and $\mathbf{b}_j$; $\mathbf{P}$ is the transportation matrix solved by the generalized Sinkhorn iteration; $\tau_1$ and $\tau_2$ determine the constraint on the reconstruction loss of pre-training and target density measures; $KL(\cdot, \cdot)$ and $h(\cdot)$ are Kullback-Leibler divergence and entropy function. Note that as a result of unbalanced total masses, we cannot perfectly reconstruct pre-training and target measures at the same time. Using this property, we can create a similarity ranking effect in the $\mathbf{P}\mathbf{1}$ vector by using a large value for $\tau_2$ but a small value for $\tau_1$. $\mathbf{P}\mathbf{1}$ is the density measure of pre-training data and $\mathbf{P}^T\mathbf{1}$ is the measure for target data. Since we want all classes of the target data to be covered, a large $\tau_2$ is needed; while we need to select a subset of classes, $\tau_1$ should be small to relax the constraint. Thus, a large $[\mathbf{P}\mathbf{1}]_j$ indicates a high similarity of class-$j$ of pre-training data to the target data. Finally by ranking the elements in $\mathbf{P}\mathbf{1}$ and selecting top-K classes, we obtain the selected classes for a target data set. Fig. 2 visualizes the UOT selection and the similarity vector given by UOT on the CUB data set.

### 3.3 Gradient Computation

We demonstrate how the gradient combination (3) is computed in the experiment of this work. In the case where pre-training data has labels, we add two classification heads on top of the network backbone. One classification head has $K_f$-dim output to predict the target data and the other has $K_g$-dim output to predict the pre-training data. The optimization objective for the labeled case is

$$\frac{1}{\tilde{n}} \sum_{t_i=1}^{\tilde{n}} f(\theta; x_{t_i}, y_{t_i}) + \frac{\lambda}{\tilde{m}} \sum_{s_i=1}^{\tilde{m}} h(\theta; x'_{s_i}, y'_{s_i}), \tag{6}$$

where $\{x_{t_i}, y_{t_i}\}$ are the target data, $\{x'_{s_i}, y'_{s_i}\}$ are the pre-training data and $\tilde{m}, \tilde{n}$ are batch size. $\lambda$ is the weight for pre-training classification loss, which controls the weight $\alpha$ in (3). Although the classification heads are different for pre-training and target data, we assume the optimization variables $\theta$ in $f$ and $h$ are consistent since the output layers only have a small amount of parameters compared to the backbone. In the case where pre-training data has no labels, the unlabeled data is used in a semi-supervised way,

$$\frac{1}{\tilde{n}} \sum_{t_i=1}^{\tilde{n}} f(\theta; x_{t_i}, y_{t_i}) + \frac{\lambda}{\tilde{m}} \sum_{s_i=1}^{\tilde{m}} h(\theta; \tilde{x}'_{s_i}, p(y_{t_i}|x'_{s_i})), \tag{7}$$

where the unlabeled pre-training data is processed with weak and strong data augmentation [31] respectively, and the probability prediction of weakly augmented data $p(y_{t_i}|x'_{s_i})$ is taken as the soft pseudo-label for the strongly augmented data $\tilde{x}'_{s_i}$. Note that there is only one classification head in (7) and temperature or threshold is not used in the weak-strong training.

## 4 Experiments

This section presents the empirical analysis of reusing pre-training data in image classification tasks. The experiment uses both supervised and self-supervised pre-trained models to fine-tune a variety of image classification data sets. First, data reusing fine-tuning schemes consistently improves the performance of vanilla fine-tuning, which corroborates our theoretical result. Second, the comparison between different data selection strategies demonstrates that the UOT selection is advantageous over random and greedy selection. Third, we simulate the situations where the training data are scarce by sub-sampling the given training data and show that as the training data get insufficient, the performance gain of the pre-training data reusing method will increase. Finally, some ablation studies on experimental settings are given.

### 4.1 Experiment Setup

The empirical study is done on both supervised and self-supervised pre-trained models. For the supervised training, we use the official ResNet18 [32] pre-trained on ImageNet. For the self-supervised training, we use the official MoCo-v2 [33] ResNet50 pre-trained with 800 epochs. In similarity-based selection, images are represented in the supervised pre-trained ResNet18 by 512-dim features from the penultimate layer while in MoCo-v2 by 128-dim features from the final

(a) Supervised Pre-Training Model

|  | Method | Dogs | Cars | CUB | Pets | SUN | Aircraft | DTD | Caltech | Avg. |
|---|---|---|---|---|---|---|---|---|---|---|
| Baseline | Fine-Tune | 82.65 | 85.87 | 75.49 | 91.40 | 58.03 | 77.62 | 70.64 | 90.11 | 78.98 |
|  | Co-Tuning [21] | 80.90 | 86.48 | 76.83 | 89.92 | _58.73_ | **79.07** | 69.69 | **93.08** | 79.34 |
| Data Selection | Random | 83.29 | 86.52 | 75.54 | 91.58 | 58.18 | 78.10 | 70.69 | 90.64 | 79.32 |
|  | Greedy-OT | _84.63_ | _86.79_ | _76.92_ | _91.66_ | 58.70 | 78.43 | _70.90_ | 90.67 | _79.84_ |
|  | UOT | **84.67** | **87.03** | **77.21** | **91.98** | **59.06** | _78.94_ | **71.17** | _91.11_ | **80.15** |

(b) Self-Supervised Pre-Training Model

|  | Method | Dogs | Cars | CUB | Pets | SUN | Aircraft | DTD | Caltech | Avg. |
|---|---|---|---|---|---|---|---|---|---|---|
| Baseline | Fine-Tune | 78.64 | **91.05** | 77.44 | 90.44 | 61.12 | 87.25 | 75.80 | 92.82 | 81.82 |
| Data Selection w/ Labels | Random | 79.87 | 90.85 | 78.82 | 91.48 | 62.42 | 88.60 | 77.34 | 93.26 | 82.83 |
|  | Greedy-OT | 79.43 | 90.89 | 78.63 | 91.27 | 62.27 | **89.40** | 76.81 | 93.36 | 82.76 |
|  | UOT | 88.14 | 90.89 | 80.98 | 93.05 | 64.76 | 89.28 | 77.45 | 93.45 | 84.75 |
| Data Selection w/o Labels | Random | 79.77 | **90.93** | 77.96 | 90.70 | 62.26 | 89.17 | 76.97 | 92.72 | 82.56 |
|  | Greedy-OT | 81.16 | 90.87 | 78.63 | **91.21** | 63.41 | 89.29 | **77.29** | 93.39 | 83.16 |
|  | UOT | **81.47** | 90.91 | **78.96** | 90.39 | **63.68** | **89.59** | 77.07 | 93.27 | **83.17** |

Table 1: Comparison of testing top-1 accuracy (%) on different data sets by fine-tuning the supervised and self-supervised pre-trained model. The proposed data selection fine-tuning consistently improves the vanilla fine-tuning, with UOT being the best method. A bold number denotes the top-1 accuracy and an underlined number denotes second best accuracy.

FC layer. The pre-trained model is tested on 8 target image classification data sets, i.e. Stanford dogs (Dogs) [34], Stanford cars (Cars) [35], Caltech-UCSD birds (CUB) [7], Oxford-IIIT Pet (Pets) [36], SUN [37], FGVC-Aircraft (Aircraft) [38], Describable Textures data set (DTD) [39] and Caltech101 (Caltech) [40]. During the fine-tuning process, both the backbone and randomly initialized classification heads are updated using SGD with Nesterov Momentum. The training epochs are fixed to be 100 in our experiment for sufficient training and the learning rate is divided by 10 at 60 and 80 epoch. Other hyperparameters like initial learning rate, weight decay and $\lambda$ are determined by grid search for all selection methods in the comparison (details in the Appendix). When there are no labels in the pre-training data, K-means clustering [41] is used to estimate the cluster assignment with 128-dim features as input and cosine similarity as distance. The cluster number is set as 2000 and we give an ablation study in Table 2 on the cluster number.

We test 3 data selection methods: random selection, greedy selection and UOT selection, and set the number of selected classes to be 100 unless mentioned otherwise. Specifically, we use the OT-based greedy algorithm [19] for comparison. The batch size for fine-tuning data is 256, and if pre-training data are reused, the batch size keeps the same as target data which makes a total batch size of 512. In random selection, we use the uniform selection over classes or clusters to be consistent with the other data selection methods. In Greedy-OT, we use the same setting as in the original paper where $\mathbf{C}_{ij}$ is the $l_2$ distance. In UOT, we set $\epsilon = 1.0$, $\tau_1 = 1.0$ and $\tau_2 = 100.0$. The distance cost is based on the cosine similarity $\mathbf{C}_{ij} = \frac{-\cos(\mathbf{a}_i, \mathbf{b}_j) + 1}{\epsilon_c}$ with $\epsilon_c = 0.01$.

For the supervised pre-trained model, we also compare with Co-Tuning [21], a strong transfer learning baseline. The experiment setting follows the original paper and we search the initial learning rate from {1e-4, 3e-4, 1e-3, 3e-3, 1e-2} on a validation set and report the test accuracy trained on the original training or train+val set. Note that Co-Tuning relies on the labels and the classification head in pre-training so it is non-trivial to use Co-Tuning for a self-supervised pre-trained model. We only compare with [21] since it is the most recent baseline which surpasses existing baselines on those datasets we evaluate. [22] needs to train another policy network to help the adaptive fine-tuning, which is more complex than ours. [19, 20] are pre-training methods that needs pre-training for every new task. [18] does not consider the pre-training and fine-tuning pipeline and needs low-level features to do data selection.

## 4.2 Comparison of Data Selection Strategies

Tab. 1 shows the comparison between the standard fine-tuning, Co-Tuning and 3 data reuse methods on 8 image classification data sets, with supervised and self-supervised pre-trained model. In the

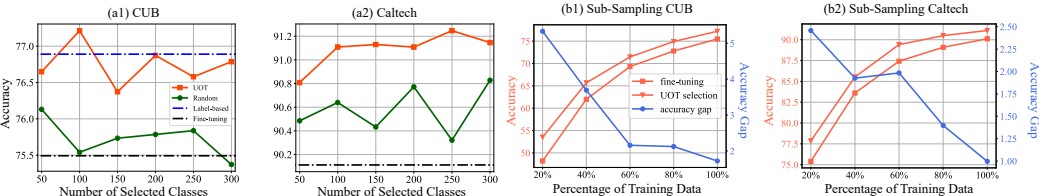

Figure 3: **(a)** Accuracy of fine-tuning using UOT data selection with different numbers of selected classes using the supervised pre-trained ResNet18. (a1) shows the performance on CUB and the blue line is fine-tuning with all birds classes from ImageNet. The UOT selection achieves a comparable performance to the label-based data selection. (a2) shows the increased performance of UOT selection on Caltech as more data are reused in UOT, while the performance of random selection is consistently worse than the UOT's. **(b)** Accuracy and performance gap when sub-sampling training data using the supervised pre-trained ResNet18. (b1) and (b2) show a decreasing trend of performance gain when more training data are added on CUB and Caltech. The advantage of pre-training data reusing is larger when training data are not sufficient.

| K | Dogs | Cars | CUB | Pets | SUN | Aircraft | DTD | Caltech | Avg. |
|---|------|------|-----|------|-----|----------|-----|---------|------|
| 1000 | 81.02 | 90.86 | 78.48 | **90.55** | 63.65 | 89.16 | **77.50** | 93.09 | 83.04 |
| 2000 | **81.47** | **90.91** | **78.96** | 90.39 | **63.68** | **89.59** | 77.07 | **93.27** | **83.17** |

Table 2: Ablation study on cluster number. The test accuracy of fine-tuning when different cluster numbers are used in the K-means algorithm demonstrates that K=2000 gives better generalization than K=1000.

labeled data, 100 classes of ImageNet data are reused. In the unlabeled data, 200 clusters are used to keep the number of selected images about the same as that in labeled data selection.

The first observation is that, since the pre-training data are large enough to have similar images to target ones, even random selection achieves better performance than the standard fine-tuning in most data sets. Secondly, the benefit of data reuse is amplified by the similarity-based data selections, as predicted by Theorem 1. Thirdly, UOT is better than Co-Tuning in 6 out of 8 data sets and its average accuracy has a clear benefit over Co-Tuning, indicating that the effectiveness of Co-Tuning is not robust to task variation. In addition, the proposed data selection is more versatile since it performs well on the self-supervised model while Co-Tuning cannot handle the self-supervised model. Finally, the comparison between Greedy-OT and UOT data selections demonstrates the advantage of the global UOT in terms of the similarity measure. Fig. 3.a1 shows the performance of label-based selection on CUB (blue line), since the birds classes happen to exist in ImageNet. It turns out the accuracy of UOT selection method (77.21%) is comparable to the label-based selection's (76.89%). In addition, we also test the performance of label-based selection on Dogs (selecting 118 dog classes of ImageNet), the performance of which (85.05%) is again comparable to UOT's (84.67%). This comparison demonstrates that the proposed UOT selection is generic yet effective.

The advantage of UOT is more evident in the self-supervised pre-training case than in the supervised pre-training one. The most considerable improvement is achieved in Dogs, Birds and Pets data sets because the animal-related classes are dominant in ImageNet (398 classes of birds, dogs, animals and mammals) and self-supervised training learns good visual features without label information. Once the label information is added to the fine-tuning process by data reuse, the model is taught to recognize those familiar features and achieves giant improvements. Note that the only data on which data reuse does not help is the Cars, indicating that the gap between ImageNet and Cars data is large when measured by the self-supervised model. When image labels in self-supervised pre-training are not available, the proposed data selection framework still outperforms the vanilla fine-tuning baseline on most data sets. The benefit of UOT over greedy-OT is large on Dogs and CUB data sets, indicating that UOT is still better at selecting similar images from ImageNet even when label information is completely unknown.

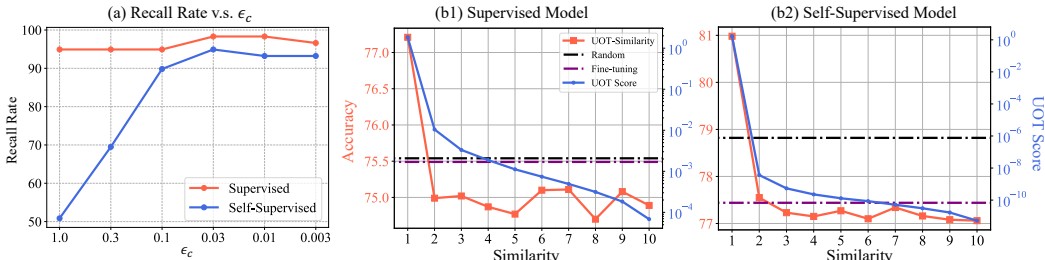

Figure 4: **(a)** Sensitivity of recall rate to $\epsilon_c$. On the supervised pre-trained model, the recall rate is not sensitive to $\epsilon_c$; on the self-supervised model, when $\epsilon_c$ is small, the performance is stable. **(b)** The similarity of selected ImageNet split to the downstream CUB data set versus downstream accuracy.

## 4.3 Simulation of Low-Data Regime and Long-Tail Label Distribution

To study the effect of data reusing in the scarce data scenario, we simulate low-data target tasks in this experiment by sub-sampling CUB and Caltech training data. We select these two data sets because they represent the fine-grained and general classification task, respectively. For each class of training data we randomly sample 20%, 40%, 60% and 80% of images to get class-balanced training data. Figure 3b shows that accuracy and performance gap between vanilla and data-reusing fine-tuning when different amounts of training data are available. On both data sets, the performance gap is increased as the training data size decreases, indicating that the UOT-selection data reusing scheme helps more when the target data is insufficient. This experiment demonstrates that the proposed data reusing paradigm is particularly effective when the target task does not have enough data, which could be a typical case in real-world applications.

The long-tail class distribution is also a challenge in real-world applications. We simulate long-tail data sets with CUB, Aircraft and Caltech by sampling images from classes with a Pareto distribution [42] as in [43], where the number of images in the largest class is 10 times that in the smallest class. The results of fine-tuning a ResNet18 are shown in the Table 3. The proposed method has a more

|  | CUB | Aircraft | Caltech |
|---|---|---|---|
| Fine-Tuning | 52.73 | 52.74 | 73.96 |
| UOT | **57.27** | **53.96** | **78.20** |

Table 3: The result of UOT-fine-tuning in data sets with long-tail class distribution.

significant improvement when the target data is imbalanced compared with the original balanced class data sets. The reason is that the selected pre-training data can be controlled to have a balanced label distribution and the gradients of pre-training data have a regularization effect, so the model does not easily overfit images in minor classes.

## 4.4 Ablation Study

**Number of selected pre-training classes.** We investigate the effect of selected class numbers on the target classification accuracy. Figure 3a shows the performance of target tasks (CUB and Caltech) when the number of selected classes ranges from 50 to 300 in UOT selection. The increased pre-training data added in fine-tuning do not improve the performance of CUB, since there are 59 classes of birds in the ImageNet and more reused images enlarge the gap $\delta^2$. Surprisingly, we observe that only using the bird images (blue line) is not the best strategy on CUB. It is because that there can be a certain number of related classes in ImageNet, which will help the classification of bird images. The result shows that even when labels of pre-training and target data are given and overlapped, UOT selection can achieve better performance by including extra relevant classes from pre-training data. On the general classification data set (Caltech), more reused pre-training data help gain the performance improvement because the diverse data set needs a large number of images to have a small domain gap. On both data sets, the UOT selection performs better than the random selection as the number of selected classes changes.

**Cluster number in K-means.** The cluster assignments are crucial to the performance of similarity-based selection, so we change the cluster number in the experiment and show the fine-tuning results of 8 data sets in Table 2. On most data sets, K=2000 achieves better generalization than K=1000,

indicating that fine-grained clustering helps the following data selection step. However, if we further increase K, the clustering result will be quite noisy and extremely small or large clusters will be found, which makes the fine-tuning result worse than K=2000.

**Distance function and $\epsilon_c$.** To investigate the influence of different factors in the UOT selection, we define a recall rate as a metric to make the comparison. For a target data set whose classes happen to exist in ImageNet, the similarity-based data selection is expected to choose those matched classes, e.g., selecting all 59 birds classes from ImageNet when fine-tuning on CUB. Thus, the recall rate on CUB is defined as the ratio between the number of bird classes in the top-100 similar vector or EMD similarity and 59. With the performance metric, we first compare UOT with Greedy-OT using $l_2$ and

| Method | Metric | GOT-$l_2$ | GOT-cos | UOT-$l_2$ | UOT-cos |
|--------|--------|-----------|---------|-----------|---------|
| Sup. | Rec. | 86.44 | 94.92 | 94.92 | **98.31** |
| | Acc. | 76.92 | 76.67 | 77.08 | **77.21** |
| Self Sup. | Rec. | 16.95 | 38.98 | 16.95 | **93.22** |
| | Acc. | 78.63 | 79.32 | 78.48 | **80.98** |

Table 4: Ablation study on distance. The UOT-cos selection is better than Greedy-OT (GOT) selection in terms of bird classes recall rate (Rec.) and test accuracy of fine-tuning (Acc.).

cosine distance in Table 4. With a supervised pre-trained model, Greedy-OT is only slightly worse than UOT, while with a self-supervised model the weakness of Greedy-OT is amplified. It means that Greedy-OT heavily relies on the label information in supervised training but UOT only needs generic visual features to have a good similarity measure. In addition, the cosine distance is better than the $l_2$ distance, especially in the self-supervised model. The importance of cosine distance is due to the cosine similarity loss used in MoCo training [33]. Finally, Fig. 4a shows the recall rate when using different $\epsilon_c$. The supervised model is not sensitive to the choice of $\epsilon_c$ but a small $\epsilon_c$ is crucial to the good performance of OT-selection in the self-supervised model. Note that the recall rate of Greedy-OT does not depend on $\epsilon_c$ so the performance is worse than UOT no matter what $\epsilon_c$ is used.

**The impact of the domain gap.** We split ImageNet into 10 subsets based on UOT score (higher means higher similarity) to CUB data and reuse the 10 subsets in fine-tuning to see the results, which are presented in Fig. 4.b1-2. With both supervised and self-supervised model, when the UOT similarity score rapidly decreases, the performance of UOT-fine-tuning drops and is lower than the fine-tuning baseline. This experiment indicates that the dissimilar data hurt the performance when added to fine-tuning and highlights the importance of similarity-based data selection.

**Subset of Pre-Training Data.** In the case where the pre-training data is larger than the available storage capacity, a subset of the pre-training data could be used. We subsample 50% images from each class of ImageNet and use the 50% ImageNet data in the supervised ResNet18 fine-tuning. On CUB and Caltech, the UOT fine-tuning achieves 77.03% and 91.00% accuracy respectively, which only drop 0.18% and 0.11% compared with using the full ImageNet. The experiment indicates that even with half of ImageNet, fine-tuning with UOT data selection is still quite effective.

## 5 Conclusion

This paper provides the generalization analysis of pre-trained models when fine-tuned on target tasks by using excess risk bound, which suggests that the pre-trained model can have little positive influence on learning from target data under certain conditions. It also shows that the performance on the target data can be improved when similar data is selected from the pre-training data for fine-tuning. Inspired by this result, a novel similarity-based selection algorithm is developed, which is evaluated on 8 data sets and shown to be effective. Our future work will further explore the data reusing strategy in other computer vision tasks. Broadly speaking, our research underpins the importance of pre-training data in downstream applications, which is neglected by current research focusing on pre-trained models, and thus has an impact in advocating the data-centric Artificial Intelligence (AI) [44]. One limitation of the current work is that sometimes the pre-training data is private (e.g., JFT-3B [45]) and not accessible. Given the power of pre-training data as unveiled by our work, we advocate that more pre-training data should be open for the better use of pre-trained models.

## Acknowledgement

This work was supported by Alibaba Group through Alibaba Research Intern Program, the Fundamental Research Funds for the Central University of China (DUT No. 82232031) and a grant from the Research Grants Council of the Hong Kong Special Administrative Region, China (Project No. CityU 11215820).

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
