# OpenReview forum: "Improved Fine-Tuning by Better Leveraging Pre-Training Data"
_NeurIPS.cc/2022/Conference — NeurIPS 2022 Accept_

### Official Review · Reviewer_51Vh · 2022-07-10

**Rating:** 5
**Confidence:** 3
**Soundness:** 3 good
**Presentation:** 3 good
**Contribution:** 2 fair

**Summary:**

This work studies the problem of using data of the pretraining task (pretraining data) for the fine-tuning stage.

They first analyze the excess risk of the target task when the pretrained model and the pretraining data  are used, and show that the proper use of pretraining data can tighten the bound of excess risks.

For selecting the proper subset of pretraining data, they propose a similarity-based selection method based on the distributional distance between pretraining classes and target data.

Their empirical results on several image-classification tasks validates the findings of the previous analysis and the several improvements of their method over the conventional fine-tuning without pretraining data.

**Questions:**

Please see the questions mentioned above.

**Limitations:**

I do not see the limitation and negative societal impact sections. Please see my mentioned limitations above. For the societal impact, the authors may consider the impact of the work on utilizing more pretraining data then standard fine-tuning.

**Strengths And Weaknesses:**

- Problem setting: The use of selected source data to improve target task has been touched before in the transfer learning literatures. This paper studies a seemly new setting: both pretrained model and pretraining data are used in the fine-tuning task. Though it’s new, my question are how important is this setting and under which scenarios is this setting useful? To be specific, I see the overlap of pretrained model and the pretraining data: If the selected pretrained data is more valuable than the pretrained model, why don’t we train the model from pretraining data and the fine-tuning data? Otherwise, as the purpose of the pretrained model is to provide an initially good (and general) representation, if the pretrained model is well-trained, it already well utilized the pretraining data. Also, as training from scratch and pretraining are usually expensive and intensive, does the use of large pretraining data contradict to the purpose of fine-tuning? (In the experiment, the paper use1 up to 100 classes of ImageNet which is approximately 100K samples while the target data (e.g., Stanford Cars) has much less data.

- Analysis: The lemma and theorem are simple but useful to illustrate the idea of this paper. From my understanding, the main contribution of these analyses is to show the usefulness of proper pretraining data selection, rather than study the defined problem in general.

- Similarity-based method:  The method is intuitive.
    - One concern is how K-mean on high-dimensional data is useful to provide distinguishable clusters (especially with Euclidean distance)? If these clusters are uniformly sampled from the data, the method is likely to be the random selection.
    - Another question I have is the justification of the real effect of pretraining data. Since the paper simply incorporate the pretraining data via extra regularization term (loss) in the fine-tuning. It’s intuitive that this regularization can at least achieve better performance than the original loss (fine-tuning) with proper regularization parameters. And, the experimental results also show the marginal improvements of the modified loss over the standard fine-tuning and sometimes gets lower performance (table 1b).

- Clarify: the paper is pretty clear to follow and easy to understand.

- Quality:
    - Writing can be improved:
        - In paper 2, the statement “From the perspective …” in lines 60—63 is mostly duplicated with lines 64—66
        - Some statements need to be provided with evidences and more precise contexts. For examples, lines 32—33: “…. training from scratch never matches the performance of fine-tuning …. “
        - Minors: there are some typos, e.g., ‘datausing’ (Figure 3 caption), ‘images’ (line 172) should be ‘samples’.

- Experiments
    - Some results seem to be  not representative to me. For example, the Fig.3a shows that 100-class setting provides the best overall performance of method while other settings result in lower performance than the label-based selection. It’ll be better to provide more comparison and robust metrics to the table 1.
    - While table 1 shows the improvement of using proposed method, the improvement is marginal and unclear whether the gain is from the pretraining data or proper parameter tuning?
        - As mentioned above, all considered target tasks have smaller number of data compared to the pretraining data. I think it’s better to show with more settings when the pretraining data is much smaller.
    - Baselines: I suggest the authors to try simple but related baselines: training both pretraining data and target data without pretrained models?
    - Some results in table 1 are far below the best transfer-learning results on the target datasets. For example, on the Stanford Cars, the latest transfer learning may achieve up to 96% accuracy.
        - What happen if we use stronger pretrained models?
    - In figure 3a-2, why don’t we have the label-based line?
    - In figure 3b, what is the accuracy gap?

---

> ### Author Response · Authors · 2022-08-02
> **Response to Reviewer 51Vh: part 1**
>
> We thank the reviewer for the careful reading and constructive feedbacks, with with we believe our updated paper is more clear and the experiment is much more convincing for readers. Here are our responses to the comments.
>
> **Q1: If the selected pretrained data is more valuable than the pretrained model, why don’t we train the model from pretraining data and the fine-tuning data?**
>
> A1: Our paper does not claim that the pre-trained data is more valuable than the pre-trained model. The analysis in Sec. 2 shows that the pre-trained model can be potentially improved by using pre-training data in the fine-tuning process. Kindly note that both the pre-trained model and a selection of the pre-training data are used in fine-tuning, thus both are important. If the pre-trained model is not used as the basis for fine-tuning, then we lose the benefit of the general representation learned from large-scale data, see the experiment in Q8.
>
> **Q2: Does the use of large pretraining data contradict to the purpose of fine-tuning?**
>
> A2: The main purpose of fine-tuning from pre-trained models is to leverage the general representation learned from large-scale data, which is useful when the downstream data is limited. The purpose of our study is to show that it is possible to improve the performance of a downstream task where the labels are limited by better leveraging the existing pre-training data. While this incurs additional cost in fine-tuning due to larger data, we also see improved downstream performance without needing to collect new downstream data.  We also note that we use the SGD with mini-batch in the training and keep the ratio between source and target batch size to be 1, so the required GPU memory is increased but the computational time keeps roughly the same (only the data sampling overhead).
>
> **Q3: How K-mean on high-dimensional data is useful to provide distinguishable clusters (especially with Euclidean distance)?**
>
> A3: On some fine-grained classification tasks (Dogs and CUB), we find that the performance of UOT is much higher than random selection, indicating that the K-means learns meaningful feature clustering.
>
> **Q4: The justification of the real effect of pretraining data.**
>
> A4: Our theoretical analysis in Sec 3.1 indicates that with such a regularization term, the performance is potentially higher than the standard fine-tuning. Specifically, Table 1a shows that the UOT fine-tuning improves upon the standard one on 7/8 datasets with an average increase 2.93% when using the ResNet50.
>
> **Q5: Writing can be improved.**
>
> A5: Thanks for your careful reading. We have revised those typos in the updated version.
>
> **Q6: Fig.3a shows that 100-class setting provides the best overall performance of method while other settings result in lower performance than the label-based selection.**
>
> A6: Fig. 3a shows the performance of using different number of classes  from ImageNet on CUB. The figue shows that the performance is always higher than other two baselines across different class numbers. Notice that the label-based selection is considered as an oracle for CUB data since we are able to select all related source data based on the label text. But as mentioned in Line 51-54, label-based selection is not as general as UOT data selection since the former depends on correspondence between labels.
>
> **Q7: It’s better to show with more settings when the pretraining data is much smaller.**
>
> A7: We simulate the small pre-training data by subsampling half of images from each class of ImageNet. On CUB and Caltech, the UOT fine-tuning achieves 77.03% and 91.00% accuracy respectively, which only drop 0.18% and 0.11% compared with using the full ImageNet. The experiment shows that even with half of ImageNet, fine-tuning with UOT data selection is still quite effective. We have added this experiment to the updated paper in Line 380-385.

---

> > ### Author Response · Authors · 2022-08-02
> > **Response to Reviewer 51Vh: part 2**
> >
> > **Q8: I suggest the authors to try simple but related baselines: training both pretraining data and target data without pretrained models.**
> >
> > A8: This is a good question. Training from scratch often takes much more time than starting from a pre-trained model. The target data size is often much smaller compared with the pre-training data. Such imbalance is an obstacle to learning the target task. Starting from a pre-trained model, the time and imbalance issue are not serious, since the pre-trained model already learns a good initialization for optimization and the model has a lower loss on the pre-training data, which will not affect learning in the target task. We run this baseline on CUB and Caltech with supervised ResNet18 and find the accuracy is 71.75% and 61.96%, which is far below the fine-tuning accuracy and indicates the importance of using the pre-trained model. We will run more experiment on other datasets and add the result later.
> >
> > **Q9: What happen if we use stronger pretrained models?**
> >
> > A9: Since a ResNet18 is used in our experiment, the performance in Table 1 does not match the state-of-the-art performance achieved by larger models. A small model is used to show the effectiveness of UOT fine-tuning in such resource-limited cases. We run the experiment using a supervised pre-trained ResNet50 and find that the UOT fine-tuning increases the accuracy of standard fine-tuning from 80.64% to 81.24% on CUB and from 88.90% to 90.82% on Dogs, indicating that the proposed method is effective for a larger supervised pre-trained model. More experiments on other datasets are still running.  We kindly note that we use a larger model (ResNet50) in self-supervised pre-training experiment, and the performance of UOT fine-tuning is quite competitive in Table 1b.
> >
> > **Q10: In figure 3a-2, why don’t we have the label-based line?**
> >
> > A10: This is a good question. Unlike CUB and Dogs, the object classes in Caltech are diverse. So the correspondence between labels of Caltech and of ImageNet cannot be trivially obtained. For datasets like Caltech, our feature-based data selection method is more appropriate than the label-based data selection.
> >
> > **Q11: In figure 3b, what is the accuracy gap?**
> >
> > A11: The accuracy gap means the accuracy of UOT fine-tuning minus standard fine-tuning.
> >
> > **Q12: The authors may consider the impact of the work on utilizing more pretraining data than standard fine-tuning.**
> >
> > A12: ImageNet is open sourced and widely used in the research community. With our UOT fine-tuning, we improve the performance of pre-trained models in target tasks, which we believe has no negative societal impact. However, our method has a limitation when the pre-training data is private (e.g., JFT-3B). As mentioned by the reviewer, our method also incurs larger computational cost than not using selected pre-training data during fine-tuning, but this cost is still much less than when training from scratch.

---

> > > ### Comment · Reviewer_51Vh · 2022-08-04
> > > **Clarification questions**
> > >
> > > Thanks the authors for the responses.
> > > I've read all the comments, but I have some concerns.
> > >
> > > 1.  (Statistically significant difference between scores) Have you computed the standard deviations of reported scores in the Table 1 and the scored reported in the your answer to Q8? If so, can you please report them. To me, the differences between methods for most of the datasets  are not enormous with respect to baseline accuracies and maybe not statistically significant.
> > >
> > > 2. (Motivation) When and why is the problem setting useful? as we need to have both pretrained models, pretraining data, and finetuning data.
> > > I think the authors suggest a use-case when the target task has very limited labeled data (If so, I believe  both types of  data needs to be relevant.)
> > > Other than that, I do not get under which scenario the setting is useful in the general finetuning setting?
> > >
> > > 3. (Significance) As mentioned  the main contribution of the analyses is to show the usefulness of proper pretraining data selection, rather than study the defined problem (showing the upper bound of the risks).
> > > The experiments only focus on very simple vision datasets with small models and small downstream tasks (and achieve lower performances than  performance), so I am not convinced whether this method really works for general cases , e.g., pretrained language models or large pretrained vision models.
> > >
> > > 4. Justification of the usefulness of pretraining data. As previously mentioned, the analyses do not directly lead to the design of the method. I would like to understand the justification and intuition why the pretraining data helps as we assume the pretrained models already captured good representation of pretraining data (overlapping between data and models).
> > >
> > > 5. For Q8, what I suggested is to use selected subset of pretraining data and the finetuning dat which should be computationally similar to your approach. The setting in Table 1 sufficiently seem doable with small models (ResNet) and small finetuning datasets. Though I trusted the results, I am not quite sure why with 100K samples +finetuning data, the ResNet18 gets  much lower performance. Have you trained in 2 phases: first pretraining data, then finetuning data?
> > >
> > > Thanks,

---

> > > > ### Author Response · Authors · 2022-08-05
> > > > **Response to Clarification questions: part 1**
> > > >
> > > > **Q13: (Statistically significant difference between scores) Have you computed the standard deviations of reported scores in the Table 1 and the scored reported in the your answer to Q8? If so, can you please report them. To me, the differences between methods for most of the datasets are not enormous with respect to baseline accuracies and maybe not statistically significant.**
> > > >
> > > > A13: Since we choose the best performance for both baselines and our method, we believe the comparison is fair. In the response to Reviewer phCA, we run 5 trials for UOT fine-tuning on CUB and Caltech with the supervised pre-trained ResNet18, the result is 76.99%±0.31% and 91.08%±0.24%, which are also clearly higher than the maximum performance of the baselines. We are running the 5-trial experiment on other datasets.
> > > >
> > > >
> > > > **Q14: (Motivation) When and why is the problem setting useful? as we need to have both pretrained models, pretraining data, and finetuning data. I think the authors suggest a use-case when the target task has very limited labeled data (If so, I believe both types of data needs to be relevant.) Other than that, I do not get under which scenario the setting is useful in the general finetuning setting?**
> > > >
> > > > A14: The problem setting is shown in Fig. 1, and the reason why this setting is useful is demonstrated by our theoretical analysis, where re-using pre-training data that is close to the downstream task will improve the generalization bound. The effectiveness of the proposed UOT fine-tuning is shown on 8 general fine-tuning data sets with both supervised and self-supervised pre-trained model. Please elaborate on which specific point is not fully understood. Are you asking how different the pre-trained data can be before it no longer is useful for fine-tuning? Note that in the updated version, we show the experiment of using different subsets of ImageNet according to the UOT similarity in Fig 4b and Line 374-379. The experiment indicates that the dissimilar data hurt the performance when added to fine-tuning and highlights the importance of similarity-based data selection.
> > > >
> > > > **Q15: (Significance) As mentioned the main contribution of the analyses is to show the usefulness of proper pretraining data selection, rather than study the defined problem (showing the upper bound of the risks).**
> > > >
> > > > A15: We do not quite understand the reviewer’s point. What is the meaning of “the usefulness of proper pretraining data selection” and “study the defined problem (showing the upper bound of the risks)”? Both Lemma 1 and Theorem 1 assume the pre-training and fine-tuning pipeline and explain the reason why pre-training data selection is effective with the upper bound of risks, see our proof in Line 667-694 of the full paper.   Empirically, we show the usefulness of using the selected pretraining data through improved performance on 8 general fine-tuning datasets with both supervised and self-superised pre-trained models.
> > > >
> > > > **Q16: The experiments only focus on very simple vision datasets with small models and small downstream tasks (and achieve lower performances than performance), so I am not convinced whether this method really works for general cases , e.g., pretrained language models or large pretrained vision models.**
> > > >
> > > > A16: As we note in the response to all reviewers, the major contribution lies in both the theoretical understanding of using pre-training data in the fine-tuning stage (Theorem 1) and the proposed UOT data selection method. Though we agree that any paper can be made better with more experiments, the current experiment has shown the effectiveness of UOT fine-tuning in both supervised and self-supervised pre-trained models on a wide range of datasets, which is acknowledged by Reviewer **718u**, **HBDj** and **phCA**.
> > > >
> > > > **Q17: Justification of the usefulness of pretraining data. As previously mentioned, the analyses do not directly lead to the design of the method. I would like to understand the justification and intuition why the pretraining data helps as we assume the pretrained models already captured good representation of pretraining data (overlapping between data and models).**
> > > >
> > > > A17: As we have claimed, the data selection approach is informed by Theorem 1. Note that though the pre-trained model learns general representations for the whole pre-trained data, not good representations for a specific downstream task. To better learn specialized features in downstream task, a similar subset of pre-training can be re-used in the downstream task. Theorem 1 states that better generalization bounds can be achieved by selecting the pre-training data that is closer to the downstream data. Thus we design our UOT selection algorithm based on feature similarity to select the pre-training subset.  Our design is corroborated by experiment in Table 1.

---

> > > > > ### Comment · Reviewer_51Vh · 2022-08-05
> > > > > **Clarification questions - Follow-up**
> > > > >
> > > > > Thanks authors for prompt responses. I appreciate the clarification.
> > > > >
> > > > > I think you might have not fully understood some of my points. So please let me clarify more clearly.
> > > > > My concerns are at two main points.
> > > > > 1. The practical use of your proposed problem. I think the problem setting is interesting but just wonder when we will benefit from collecting more pretraining data (and in which case they are also available) and trade-off this cost of collecting+training with small increases of accuracy? that's what I mean "under which scenarios" (Q14) . This leads to my second question (Q16) -- if we can show that this setting is widely applicable then it's great.
> > > > >
> > > > > 2. Sufficiency of evidence for the effectiveness of the proposed method. Please correct me if I did not understand correctly, but as I understood, the theorem is to inspire your method, not the guarantee of the effectiveness of your method (my question Q17).
> > > > > - Thus, the empirical evidence needs to be sufficient and representative for the general finetuning setting. The datasets are all object-classification vision datasets. For Q16, I don't mean to have huge-dataset experiment but more representative as your proposed method doesn't limit the scope to only vision.  I believe that similar simple experiments but in other modalities (rather than vision) will make your message more convincing.
> > > > > - Otherwise, if I understood correctly, the authors meant to frame the contribution of the paper as both theory and methodology. While they are coherent and the message is solid, the contribution of each end seems to be less significant.
> > > > >
> > > > > For Q13, the reason I asked for std because (1) randomness may have huge effect in the result, and (2) the results your method achieves are not beyond the SOTA reported by others. Here, I do not mean to achieve SOTA, but because of (2), (1) can take place and can become a factor to the gap of best models.

---

> > > > > > ### Author Response · Authors · 2022-08-05
> > > > > > **Response to Follow-Up: part 1**
> > > > > >
> > > > > > Thanks for your clarification. Here is a quick response.
> > > > > >
> > > > > > **The practical use of your proposed problem. I think the problem setting is interesting but just wonder when we will benefit from collecting more pretraining data (and in which case they are also available) and trade-off this cost of collecting+training with small increases of accuracy? that's what I mean "under which scenarios" (Q14) . This leads to my second question (Q16) -- if we can show that this setting is widely applicable then it's great.**
> > > > > >
> > > > > > A: Again, we have shown that the method is effective in a wide range of image classification datasets. And the correlation between similarity and performance has been shown in Figure 4b in the updated paper, which shows a higher similarity leads to a better performance in general.
> > > > > >
> > > > > > **Sufficiency of evidence for the effectiveness of the proposed method. Please correct me if I did not understand correctly, but as I understood, the theorem is to inspire your method, not the guarantee of the effectiveness of your method (my question Q17).**
> > > > > >
> > > > > > A: The theorem guarantees that the generalization upper bound is reduced if the gap between selected source and target data is small, which directly leads to the pre-training data selection and the algorithm. Note that the delta in Theorem 1 is the maximum distance between input gradients of source and target data, with the UOT data selection, the distance should be smaller than without the selection, meaning that the in theory the UOT method would be not worse that the reuse method of pre-training data WITHOUT selection. Since the $\delta$ in Theorem 1 could be arbitrary large, the excess risk bound could be very large, which leads to worse performance of the naive reuse method without selection. However, by contrast,  the UOT method could perform better by carefully controlling $\delta$. Empirically, Figure 2 in the updated version shows that UOT learns a quite good similarity measure because most bird classes (58 out of 59) from ImageNet are selected in the top-100 similarity vector to CUB. Could the reviewer explain what is the meaning of “the guarantee of the effectiveness of your method” by giving an example? Does the reviewer mean the proof for the effectiveness of UOT? Since the current paper focuses on the improvement of fine-tuning, a theoretical contribution on optimal transport theory is out of the scope of our current paper.
> > > > > >
> > > > > >
> > > > > > **Thus, the empirical evidence needs to be sufficient and representative for the general finetuning setting. The datasets are all object-classification vision datasets. For Q16, I don't mean to have huge-dataset experiment but more representative as your proposed method doesn't limit the scope to only vision. I believe that similar simple experiments but in other modalities (rather than vision) will make your message more convincing.**
> > > > > >
> > > > > >
> > > > > > A: As we have responded to Reviewer phCA, we agree that NLP experiments are also interesting to explore, but due to limited space, the current paper focuses on the theoretical analysis and the fine-tuning of supervised and self-supervised computer vision models with “solid ablation study” (by Reviewer phCA). We notice that there are already some NLP papers proposing to retrieve data for better pre-training and fine-tuning, e.g. [R3,R4], and our analysis has the potential to provide a theoretical justification for using pre-training in NLP. We will try our best to run an NLP experiment, but given the limited time it is not guaranteed to be finished within the discussion period.
> > > > > >
> > > > > > [R3] Guu, K., Lee, K., Tung, Z., Pasupat, P., & Chang, M. (2020, November). Retrieval augmented language model pre-training. In International Conference on Machine Learning (pp. 3929-3938). PMLR.
> > > > > >
> > > > > > [R4] Khandelwal, U., Levy, O., Jurafsky, D., Zettlemoyer, L., & Lewis, M. (2019, September). Generalization through Memorization: Nearest Neighbor Language Models. In International Conference on Learning Representations.
> > > > > >
> > > > > > **Otherwise, if I understood correctly, the authors meant to frame the contribution of the paper as both theory and methodology. While they are coherent and the message is solid, the contribution of each end seems to be less significant.**
> > > > > >
> > > > > >
> > > > > > A: In the theory part, the theoretical contribution explains why and when re-using pre-training data in the fine-tuning can be effective, which has not been studied in existing literature and has a substantial contribution. In the methodology part, the UOT data selection has been shown to be effective on 8 vision datasets, with not only the supervised model, but also the self-supervised model, which again has not been investigated before. Thus, we believe the contributions in both ends are significant.

---

> > > > > > ### Author Response · Authors · 2022-08-05
> > > > > > **Response to Follow-Up: part 2**
> > > > > >
> > > > > > **For Q13, the reason I asked for std because (1) randomness may have huge effect in the result, and (2) the results your method achieves are not beyond the SOTA reported by others. Here, I do not mean to achieve SOTA, but because of (2), (1) can take place and can become a factor to the gap of best models.**
> > > > > >
> > > > > >
> > > > > > A: As we have shown in A13, the standard deviation is not huge in our understanding. We would provide the standard deviation of other datasets when the experiment is finished.
> > > > > >
> > > > > > Hope our clarification will address your concerns.

---

> > > > > > > ### Comment · Reviewer_51Vh · 2022-08-07
> > > > > > > **Re: Response to Follow-Up**
> > > > > > >
> > > > > > >
> > > > > > > Thanks the author for the responses.
> > > > > > >
> > > > > > > Again, for the "The practical use of your proposed problem" , I don' think you answered my  very first question. I was questioning that "giving both pretraining data and pretrained models" is a strong assumption and maybe unrealistic in practice whether people likely to use. I was trying to see if you found any scenarios to motivate the use. I do not mean 'vision' or 'nlp'. Sorry if my question confuses you.
> > > > > > >
> > > > > > > I appreciate the responses and the soundness of work and i find it interesting. However, relatively compared to other papers in Neurips, i cannot see much significant contributions and impacts of this work to the ML community, in research methodology and practical use.
> > > > > > > Nevertheless, I am willing to increase my score to 5 as if the updated stds from the authors show that the improvements in table 1 is statistically significant.
> > > > > > > Best,

---

> > > > > > > > ### Author Response · Authors · 2022-08-08
> > > > > > > > **Re: Re: Response to Follow-Up**
> > > > > > > >
> > > > > > > > We thank the reviewer for providing valuable feedbacks during the discussion phase so that we have the opportunity to address concerns and clarify some key issues. It is greatly appreciated that the reviewer agrees to raise the score if we provide the statistical test for the effectiveness of UOT. Here are our response and new experimental results.
> > > > > > > >
> > > > > > > > * Though we admit that there exist certain pre-trained datasets that are not publicly available as mentioned in the Conclusion section, most widely used pre-trained datasets such as ImageNet (for vision models like ResNet, ViT, etc), Wikipedia and C4 (for NLP models like BERT, T5, etc) are readily accessible. Moreover, using such pre-training data with a pre-trained model to improve fine-tuning has been investigated by published papers, such as Co-Tuning [R5] (NeurIPS) for vision applications and [R3,R4] (ICML and ICLR) for NLP applications.
> > > > > > > >
> > > > > > > > [R3] Guu, K., Lee, K., Tung, Z., Pasupat, P., & Chang, M. (2020, November). Retrieval augmented language model pre-training. In International Conference on Machine Learning (pp. 3929-3938). PMLR.
> > > > > > > >
> > > > > > > > [R4] Khandelwal, U., Levy, O., Jurafsky, D., Zettlemoyer, L., & Lewis, M. (2019, September). Generalization through Memorization: Nearest Neighbor Language Models. In International Conference on Learning Representations.
> > > > > > > >
> > > > > > > > [R5] You, Kaichao, et al. "Co-tuning for transfer learning." Advances in Neural Information Processing Systems 33 (2020): 17236-17246.
> > > > > > > >
> > > > > > > >
> > > > > > > > * We implement the 2-phase training using a supervised ResNet18 on CUB and Caltech. The accuracy is 74.21% and 84.44% respectively, which is higher than the training from scratch result 71.75% and 61.96% but still worse than the standard fine-tuning (75.49% and 90.11%) and our improved UOT result (77.21% and 91.11%). We think the reason is that the pre-training on closely related source data has the risk of overfitting and cannot learn a general feature representation that provides a good initialization for various downstream tasks and alleviates the risk of overfitting.
> > > > > > > >
> > > > > > > > * Here is the table for UOT fine-tuning and standard fine-tuning with mean and std of 5 trials using supervised ResNet18. We report the p value of unpaired t test for the 8 datasets: on all datasets the benefit of UOT fine-tuning is statistical significant.
> > > > > > > > | Dataset  | Dogs | Cars | CUB | Pets | SUN | Aircraft | DTD | Caltech |
> > > > > > > > |----------|------|------|-----|------|-----|----------|-----|---------|
> > > > > > > > | Standard |   82.59±0.21   |  85.80±0.12    |   75.22±0.21  |   90.85±0.41   |  58.22±0.24   |      77.11±0.43    |  69.95±0.58   |    89.07±0.69     |
> > > > > > > > | UOT      |   84.59±0.22   |   86.96±0.11   |   76.99±0.31  |   91.71±0.20       | 58.78±0.22  | 78.15±0.46   |   70.85±0.29  |  91.08±0.24  |
> > > > > > > > | p value  |    <0.0001   |   <0.0001   |  <0.0001   |  0.0015    | 0.0025     |  0.0031    |  0.0073   |     0.0001    |
> > > > > > > >
> > > > > > > > Thanks again for your patience and valuable feedbacks.

---

> > > > > > > > > ### Comment · Reviewer_51Vh · 2022-08-09
> > > > > > > > > **Update score to 5**
> > > > > > > > >
> > > > > > > > > Thanks the authors for the response.

---

> > > > ### Author Response · Authors · 2022-08-05
> > > > **Response to Clarification questions: part 2**
> > > >
> > > > **Q18: For Q8, what I suggested is to use selected subset of pretraining data and the finetuning dat which should be computationally similar to your approach. The setting in Table 1 sufficiently seem doable with small models (ResNet) and small finetuning datasets. Though I trusted the results, I am not quite sure why with 100K samples +finetuning data, the ResNet18 gets much lower performance. Have you trained in 2 phases: first pretraining data, then finetuning data?**
> > > >
> > > > A18: We did not use the 2-phase training, since the 2-phase setting has already been investigated in [R1,R2] and is not the problem setting we are studying in this paper. We did not compare ours with [R1,R2], since they need to re-pre-train the model for every new dataset, which requires more time and efforts than our fine-tuning approach, see Line 95-100. We are running the 2-phase experiment now on several datasets and will report them when finished.
> > > >
> > > > [R1] Cui, Yin, et al. "Large scale fine-grained categorization and domain-specific transfer learning." Proceedings of the IEEE conference on computer vision and pattern recognition. 2018.
> > > >
> > > > [R2] Chakraborty, Shuvam, et al. "Efficient conditional pre-training for transfer learning." Proceedings of the IEEE/CVF Conference on Computer Vision and Pattern Recognition. 2022.

---

### Official Review · Reviewer_phCA · 2022-07-11

**Rating:** 5
**Confidence:** 2
**Soundness:** 3 good
**Presentation:** 3 good
**Contribution:** 2 fair

**Summary:**

The paper shows how leveraging some pre-training data during fine-tuning can boost a model's pre-train + fine-tuned performance.
The authors suggest a few methods to this end, for both the scenario when pre-training data is unlabeled (i.e. self-supervised learning) as well as labeled -- selecting pre-training data by label, randomly, and via similarity with fine-tuning data by way of optimal transport. The latter is the most innovative and performs the best. With several key assumptions (the most important of which is the bound on the distance between the fine-tuning and pre-training loss gradients), the paper provides bounds for the excess risk under normal fine-tuning as well as under their proposed modified fine-tuning loss. They compare against a single recent baseline (Co-Tuning) and study image classification on 8 datasets using ResNets pre-trained on ImageNet and do several ablation studies.

**Questions:**

Would it be possible to add at least 1-2 NLP experiments as well? Perhaps the same one used in the Co-Tuning paper?

**Limitations:**

The authors do mention limitations in the Conclusion section, which I felt was adequate. However, more discussion around the assumptions made for the theory would be nice -- how strong are the assumptions? Do they hold in practice?

**Strengths And Weaknesses:**

Strengths:
* Experiments have decent coverage (self-supervised + supervised pre-training ; limited data; 8 datasets)
* Pretty solid ablation studies
* Proposed method with UOT selection generally outperforms the other methods.
* Excess risk bound shown seems semi interesting.

Weaknesses:
* Novelty is limited -- while I cannot comment extensively of how noteworthy the theoretical analysis is, the suggested approaches for selecting pre-training data seem straightforward.
* Only co-tuning is compared against (though this method is claimed to be the most competitive recently) and only in the supervised setting. The proposed method with UOT selection loses to co-tuning on 2/8 datasets. Performance is measured as "the best among 3 trials" -- running this for more trials and reporting error bars would make the trends more convincing.
* More experiments would strengthen the paper.

---

> ### Author Response · Authors · 2022-08-02
> **Response to Reviewer phCA**
>
> We thank the reviewer for the insightful comments. We are glad that the reviewer considers our experiment and ablation study to be solid and extensive, and appriciate the novelty of the excess risk bound and the effectiveness of the proposed approach. Here are the responses to your comments.
>
> **Q1: Novelty is limited.**
>
> A1: See the comment for the novelty issue.
>
> **Q2: Running this for more trials and reporting error bars would make the trends more convincing.**
>
> A2: Since we choose the best performance for both baselines and our method, we believe the comparison is fair. We run 5 trials for UOT fine-tuning on CUB and Caltech with the supervised pre-trained ResNet18, the result is 76.99%±0.31% and 91.08%±0.24%, which are also clearly higher than the maximum performance of standard fine-tuning.
>
>
> **Q3: Would it be possible to add at least 1-2 NLP experiments as well?**
>
> A3: Thanks for mentioning NLP experiments. We agree that NLP experiments are also interesting to explore, but due to limited space, the current paper focuses on the theoretical analysis and the fine-tuning and analysis of supervised and self-supervised computer vision models.  However, we believe that the theoretical analysis is quite general and may also hold for NLP data. We notice that there are already some NLP papers proposing to retrieve data for better pre-training and fine-tuning, e.g. [R1,R2], and our analysis has the potential to provide a theoretical justification for using pre-training in NLP. We will work towards this direction in the future.
>
> [R1] Guu, K., Lee, K., Tung, Z., Pasupat, P., & Chang, M. (2020, November). Retrieval augmented language model pre-training. In International Conference on Machine Learning (pp. 3929-3938). PMLR.
>
> [R2] Khandelwal, U., Levy, O., Jurafsky, D., Zettlemoyer, L., & Lewis, M. (2019, September). Generalization through Memorization: Nearest Neighbor Language Models. In International Conference on Learning Representations.
>
> **Q4: How strong are the assumptions? Do they hold in practice?**
>
> A4: We have four assumptions for the main assumption as stated in the Appendix B. The Assumptions 1,2 and 3 generally hold for neural network functions, which are non-convex. The PL condition has been theoretically and empirically observed in training deep neural networks [R3,R4]. It is widely used to establish convergence in the literature of non-convex optimization, please see [R5,R6,R7,R8] and references therein.
>
>
> [R3] Allen-Zhu, Zeyuan, Yuanzhi Li, and Zhao Song. "A convergence theory for deep learning via over-parameterization." International Conference on Machine Learning. PMLR, 2019.
>
> [R4] Yuan, Zhuoning, et al. "Stagewise training accelerates convergence of testing error over SGD." Advances in Neural Information Processing Systems 32 (2019).
>
> [R5] Wang, Zhe, et al. "Spiderboost and momentum: Faster variance reduction algorithms." Advances in Neural Information Processing Systems 32 (2019).
>
> [R6] Karimi, Hamed, Julie Nutini, and Mark Schmidt. "Linear convergence of gradient and proximal-gradient methods under the polyak-łojasiewicz condition." Joint European conference on machine learning and knowledge discovery in databases. Springer, Cham, 2016.
>
> [R7] Li, Zhize, and Jian Li. "A simple proximal stochastic gradient method for nonsmooth nonconvex optimization." Advances in neural information processing systems 31 (2018).
>
> [R8] Charles, Zachary, and Dimitris Papailiopoulos. "Stability and generalization of learning algorithms that converge to global optima." International Conference on Machine Learning. PMLR, 2018.

---

> ### Author Response · Authors · 2022-08-08
> **Response to Reviewer phCA: Experiment of More Trials**
>
> Thanks again for your time and effort in handling our paper. We would like to note that we have an experiment with 5 trials during the discussion phase.
>
> Here is the table for UOT fine-tuning and standard fine-tuning with mean and std of 5 trials using supervised ResNet18. We report the p value of unpaired t test for the 8 datasets: on all datasets the benefit of UOT fine-tuning is statistical significant.
> | Dataset  | Dogs | Cars | CUB | Pets | SUN | Aircraft | DTD | Caltech |
> |----------|------|------|-----|------|-----|----------|-----|---------|
> | Standard |   82.59±0.21   |  85.80±0.12    |   75.22±0.21  |   90.85±0.41   |  58.22±0.24   |      77.11±0.43    |  69.95±0.58   |    89.07±0.69     |
> | UOT      |   84.59±0.22   |   86.96±0.11   |   76.99±0.31  |   91.71±0.20       | 58.78±0.22  | 78.15±0.46   |   70.85±0.29  |  91.08±0.24  |
> | p value  |    <0.0001   |   <0.0001   |  <0.0001   |  0.0015    | 0.0025     |  0.0031    |  0.0073   |     0.0001    |
>
> We hope our responses and clarifications address your concerns.

---

### Official Review · Reviewer_HBDj · 2022-07-12

**Rating:** 5
**Confidence:** 3
**Soundness:** 3 good
**Presentation:** 2 fair
**Contribution:** 2 fair

**Summary:**

Pretrained models are widely adopted for various downstream applications and have been demonstrating superior performance. A de-facto technique to utilize such a model is simply fine-tuning the pretrained model to a small downstream task’s dataset. This paper proposes a way to improve this process by selecting a subset of relevant pretraining data and training together with the downstream training data.

**Questions:**

Q1: Page 6: Is there any reason why the authors chose a small pretrained model? ResNet18 is a bit too small in my opinion. Did they consider larger pretrained models? (e.g. ResNet50 for supervised training or even ResNet101). Is it because larger models may memorize pretraining dataset better and may not benefit from using them?

Q2: Since one of the motivations of using the pretraining data is to prevent “catastrophic forgetting”, I wonder if the authors compared their work to such techniques for example [1,2]. [1] shows it’s much better than simple fine-tuning and contains many baselines on this end. [2] actively prevents catastrophic forgetting even without data.

[2] Hayes, Tyler L., et al. "Remind your neural network to prevent catastrophic forgetting." European Conference on Computer Vision. Springer, Cham, 2020.
[3] Chen, Sanyuan, et al. "Recall and learn: Fine-tuning deep pretrained language models with less forgetting." arXiv preprint arXiv:2004.12651 (2020).


**Limitations:**

Commonly, we do not have access to the pretraining data because of various reasons (c.f. access, storage space, time etc). This limits the use of this method in such scenarios.

**Strengths And Weaknesses:**

Strength
* Employing UOT (unbalanced optimal transport) to select the subset of the pretraining data is novel and was very effective from their empirical evaluation.
* It is quite surprising that utilizing the pretraining data can improve the fine-tuning performance this much.

Weaknesses
* Selecting the subset of the pretraining data for the downstream task is not novel (as described in line 92.
* Authors do not have many baseline methods in their empirical evaluation other than very simple ones. I encourage they compare their work to comparable methods such as [1] or methods that actively prevents catastrophic forgetting [2,3] (3 was not evaluated on the vision but in a similar fashion)
* (minor) as noted in the limitation, sometimes we do not have the access to the pretraining data. In the contrary [3] does work without the pretraining data.

[1] Chakraborty, Shuvam, et al. "Efficient conditional pre-training for transfer learning." Proceedings of the IEEE/CVF Conference on Computer Vision and Pattern Recognition. 2022.
[2] Hayes, Tyler L., et al. "Remind your neural network to prevent catastrophic forgetting." European Conference on Computer Vision. Springer, Cham, 2020.
[3] Chen, Sanyuan, et al. "Recall and learn: Fine-tuning deep pretrained language models with less forgetting." arXiv preprint arXiv:2004.12651 (2020).

---

> ### Author Response · Authors · 2022-08-02
> **Response to Reviewer HBDj**
>
> Thanks for your constructive comments on our paper, and noting the novelty of our UOT data selection and its effectiveness. Here are our responses to your comments.
>
> **Q1: Selecting the subset of the pretraining data for the downstream task is not novel.**
>
> A1: See the response for novelty.
>
> **Q2: Authors do not have many baseline methods in their empirical evaluation other than very simple ones.**
>
> A2: We compare with a recent improved fine-tuning baseline Co-Tuning in the supervised pre-training setting. In self-supervised pre-training, we only compare with random and greedy-OT selection since there are no fine-tuning baselines targeting self-supervised training as far as we know. As we note in Line 95-100 and Line 278-280, [R1] is a conditional pre-training method instead of a fine-tuning method, so we believe the comparison is not fair.
>
> **Q3: Sometimes we do not have the access to the pretraining data.**
>
> A3: It is a good question. The assumption that the pre-training data are readily available during the fine-tuning is generally satisfied for a wide range of computer vision tasks, since the most commonly used pre-training data ImageNet is free for non-commercial use. The disk usage of the ImageNet is ~150 GB, which is affordable for common servers. In the case where the pre-training data is larger than the available storage capacity, a subset of the pre-training data could be used. We subsample 50% images from each class of ImageNet and use the 50% ImageNet data in the supervised ResNet18 fine-tuning. On CUB and Caltech, the UOT fine-tuning achieves 77.03% and 91.00% accuracy respectively, which only drop 0.18% and 0.11% compared with using the full ImageNet. The experiment indicates that even with half of ImageNet, fine-tuning with UOT data selection is still quite effective.  We agree that availability of the pre-training is a limitation, especially for private datasets such as JFT-3B, so we have added this to the limitation.
>
> **Q4: ResNet18 is a bit too small. Did they consider larger pretrained models?**
>
> A4: ResNet18 is often used in cases where memory and/or computation are limited. We use a small model to show the effectiveness of UOT fine-tuning in such resource-limited cases. We run the experiment using a supervised pre-trained ResNet50 and find that the UOT fine-tuning increases the accuracy of standard fine-tuning from 80.64% to 81.24% on CUB and from 88.90% to 90.82% on Dogs, indicating that the proposed method is effective for a larger supervised pre-trained model. More experiments on other datasets are still running.  We kindly note that we use a larger model (ResNet50) in self-supervised pre-training experiment, and the performance of UOT fine-tuning is quite competitive in Table 1b.
>
> **Q5: One of the motivations of using the pretraining data is to prevent “catastrophic forgetting”?**
>
> A5: We kindly note that preventing catastrophic forgetting is not one of the motivations. We mention catastrophic forgetting in the comment of Lemma 1 to indicate the possibility that our analysis shows the possible connection between Lemma 1 and catastrophic forgetting. Our experiment aims to improve the performance of the target task, instead of combatting the forgetting of pre-training data. We deleted this sentence to prevent misunderstanding.
>
>
> [R1] Chakraborty, Shuvam, et al. "Efficient conditional pre-training for transfer learning." Proceedings of the IEEE/CVF Conference on Computer Vision and Pattern Recognition. 2022.

---

> > ### Comment · Reviewer_HBDj · 2022-08-09
> > **A question on catastrophic forgetting**
> >
> > Authors discuss that a fine-tuning phase usually does not have enough data for the lengthy training (line 149); hence, the use of pretraining helps to have better generalization because it's not overly fitting to the given task. From my understanding on this, the additional pre-training data plays a role of regularization and thus helping the model to perform better in the end. This feels a very similar argument of "Catastrophic failure" cases because the work also suggest to prevent model to move too far from the pretrained solution (generic) and will not be overly optimized to the downstream task.
> >
> > Since authors deny this connection, I wonder what the main reason of this improvements using pre-training data is that authors suggest? Can you discuss on this point?

---

> > > ### Author Response · Authors · 2022-08-10
> > > **Response: A question on catastrophic forgetting**
> > >
> > > Thanks for your insightful comment. Here is our response.
> > >
> > > In the data reusing framework, the random data selection plays the role of constraining the model to be close to the initialization (pre-trained model), in that it uniformly samples images from the pre-training data and aims to maintain the generic solution. In contrast, the similarity-based data selection such as UOT reuses task-related data to enrich the downstream training data and guides the pre-trained model to learn _task-specific_ representations, instead of maintaining the generic feature representation of the pre-trained model. Our experiment in Table 1 has shown that learning task-specific features (UOT) is better than keeping the general features (Random) in downstream task on a variety of datasets, unveiling the fundamental difference between our work and catastrophic forgetting.
> > >
> > > We kindly note that we do not deny the potential connection between Lemma 1 and catastrophic forgetting, but delete the sentence in case it raises the misconception that our work is motivated by catastrophic forgetting. The reviewer’s insightful question helps us better clarify the difference between ours and catastrophic forgetting and we will add the above discussion to an updated version.
> > >
> > > It would be deeply appreciated if our response could address the concerns and the reviewer could raise the score. We are willing to answer follow-up questions if the discussion does not fully address your concern.

---

### Official Review · Reviewer_718u · 2022-07-17

**Rating:** 6
**Confidence:** 4
**Soundness:** 3 good
**Presentation:** 3 good
**Contribution:** 3 good

**Summary:**

The authors investigate whether a pre-trained model can be enhanced when pre-training data are used during fine-tuning. They develop an approach that performs data selection to choose an appropriate portion of pre-training data to enhance the phase of fine-tuning. Specifically, the authors propose a selection algorithm to obtain a subset from pre-training data closest to the target data by solving an unbalanced optimal transport (UOT) problem, which chooses data classes from the pre-training set whose distributional distance to the target data set is small.


**Questions:**

- What if the training data is a duplicate of the target data? The training data may not further help the target task. For example, CUB has an overlap with ImageNet.

- What if the target label and the source label are semantically similar but not exactly similar. For example, if the target task is to classify dogs vs cats, but the source data have a lot of dog species but not the “dogs” label?

- Fine-tuning is sensitive to hyperparameters [1]. In the paper, the authors performed a large-scale grid search of the optimal hyperparameters for the best performance (5x3x3x3=145 trials per dataset). Is the method sensitive to hyperparameter changes? I am curious if the authors have any observation of whether the hyperparameters are correlated with the similarities of added source data and target data so that the HPO trials can be reduced.

- There are also methods that directly select the data that can benefit the target task (e.g., [2]), which were not discussed or compared.

[1] Li et al, Rethinking the Hyperparameters for Fine-tuning, ICLR 2020
[2] Dukler et al, DIVA: Dataset Derivative of a Learning Task, ICLR 2022

**Ethics Review Area:**

["I don’t know"]

**Limitations:**

The limitation is not discussed.

**Strengths And Weaknesses:**

Strength:

The paper is well written. The author provides both theoretical and empirical investigations. The proposed selection method considers to handle pre-training data with and without labels. The paper has an extensive comparison with other strategies.

Weakness:

- The reason why source data can help for target tasks is not well understood.
- The theoretical study does not provide clear insights on how similar the source datasets affect the fine-tuning performance.
- It is not clear whether the target task can always benefit from the techniques. The author discussed “the impact of the domain gap”, however, there is no clear criteria whether the method should be used. For example, will a medical dataset benefit from the ImageNet pre-training data?
- There are also methods that directly select the data that can benefit the target task, which are not discussed.

---

> ### Author Response · Authors · 2022-08-02
> **Response to Reviewer 718u: part 1**
>
> Thanks for your valuable comments and appriciating our strength in both theory and experiment. Here are our responses for your comments.
>
> **Q1: The reason why source data can help for target tasks is not well understood.**
>
> A1: We have a theoretical analysis on the generalization error of fine-tuning with pre-training (source) data culminating in Theorem 1, which shows that the generalization error can be further reduced if we select source data to control the gap between the reused source and downstream data during fine-tuning. This theoretical result explains why we need the source data and corroborates our intuition that the abundant image-label pairs from the pre-training data can be useful if an appropriate subset is selected.
>
> **Q2: The insights on how similar the source datasets affect the fine-tuning performance.**
>
> A2: The delta in Theorem 1 means the gap between the source and target data, measured by the difference of loss gradients between the reused source (pre-training) and target (downstream) data. A higher similarity leads to a smaller delta and a lower generalization error bound. As discussed in Line 374-379, our experiment in Fig. 4b corroborates this theoretical finding.
>
>
> **Q3: There is no clear criteria whether the method should be used.**
>
> A3: This is a great question. We use 8 datasets in our experiment including various objects and find that the UOT fine-tuning improves the performance in almost all datasets, indicating that the proposed method is effective in most general image classification tasks. However, for highly specialized tasks such as medical diagnosis (where the image domain is significantly different, e.g., microscope images or CT scans), the reusing the pre-training data from general visual recognition tasks such as ImageNet may not be very helpful. Indeed from Theorem 1, the generalization bound depends on the “delta” between the source and target data.  We will try to extend our work to medical data in the future.
>
> **Q4: There are also methods that directly select the data that can benefit the target task.**
>
> A4: In Line 89-107, we discuss several methods for data selection including [18,19,20,21]. We compare with the Co-Tuning in our experiment and discuss the relationship between our work and [18,19,20,21,22] in Line 276-280.
>
> [18] Weifeng Ge and Yizhou Yu. Borrowing treasures from the wealthy: Deep transfer learning through selective joint fine-tuning. In Proceedings of the IEEE conference on computer vision and pattern recognition, pages 1086–1095, 2017.
>
> [19] Yin Cui, Yang Song, Chen Sun, Andrew Howard, and Serge Belongie. Large scale fine-grained categorization and domain-specific transfer learning. In Proceedings of the IEEE conference on computer vision and pattern recognition, pages 4109–4118, 2018.
>
> [20] Shuvam Chakraborty, Burak Uzkent, Kumar Ayush, Kumar Tanmay, Evan Sheehan, and Stefano Ermon. Efficient conditional pre-training for transfer learning. arXiv preprint arXiv:2011.10231, 2020.
>
> [21] Kaichao You, Zhi Kou, Mingsheng Long, and Jianmin Wang. Co-tuning for transfer learning. Advances in Neural Information Processing Systems, 33, 2020.
>
> [22] Yunhui Guo, Honghui Shi, Abhishek Kumar, Kristen Grauman, Tajana Rosing, and Rogerio Feris. Spottune: transfer learning through adaptive fine-tuning. In Proceedings of the IEEE/CVF Conference on Computer Vision and Pattern Recognition, pages 4805–4814, 2019.
>
>
> **Q5: What if the training data is a duplicate of the target data?**
>
> A5: If the pre-training data is a duplicate of the target data, then the performance on the target data would be quite strong since the pre-training model generally has a quite low error on the pre-training data. We kindly note that the CUB’s classes overlap with some of ImageNet’s, but the image data are different. Nonetheless, our UOT selection method works as expected on CUB (see Table 1).
>
> **Q6: What if the target label and the source label are semantically similar but not exactly similar. For example, if the target task is to classify dogs vs cats, but the source data have a lot of dog species but not the “dogs” label?**
>
> A6: This is a great question. This is a problem of label-based data selection and we solve this problem by proposing the UOT selection, where the selection is based on pre-trained features (semantics) instead of label text, see Line 51-54.
>
> **Q7: Is the method sensitive to hyperparameter changes? Whether the hyperparameters are correlated with the similarities of added source data?**
>
> A7: Yes, we also observe that the performance is sensitive to the hyperparameters. We attached the hyperparameter result in supplemental materials. We find that on CUB, where the similarity between source and target data is small, a larger lambda generally leads to a better result, indicating that the selected data are quite important for the improved performance.

---

> > ### Author Response · Authors · 2022-08-02
> > **Response to Reviewer 718u: part 2**
> >
> > **Q8: There are also methods that directly select the data that can benefit the target task, such as [R2].**
> >
> > A8: [R2] proposes to optimize a weight for the target data samples instead of selecting pre-training data for improving fine-tuning. Such target-weighting methods are compatible with ours, which selects source data, and future work will consider their combination. We added the discussion to the updated version of our paper (Line 105-107) and cited the mentioned work.
> >
> > [R2] Dukler et al, DIVA: Dataset Derivative of a Learning Task, ICLR 2022
> >
> > **Q9: The limitation is not discussed.**
> >
> > A9: The limitation of our method when the pre-training data are not accessible is added in Line 393-396.

---

### Author Response · Authors · 2022-08-02
**Response to all reviews**

We thank all reviewers for your insightful comments on our paper, based on which we have revised our paper with the following major changes to address your concerns:

* Reviewer **51Vh** and **HBDj**: We add the ablation study of using half of ImageNet in Line 380-385, which shows that even with 50% of ImageNet the UOT fine-tuning is still effective.

* Reviewer **51Vh**: We add the baseline of training from scratch with selected source and target data in Line 615-623 of the full paper in supplemental. The experiment shows the importance of using pre-trained model in the fine-tuning phase.

* Reviewer **718u**, **HBDj** and **51Vh**: The limitation of our method when the pre-training data are not accessible is discussed in Line 393-396.

* Reviewer **718u**: We added the hyperparameter search result in the supplemental material (UOT_hyper_search.xlsx).

* Reviewer **phCA**: We run 5 trials of UOT fine-tuning and add the result to the full version (Line 581-584) and discussed the assumptions in Line 641-642.

* Reviewer **51Vh**: The mentioned typos are fixed in the updated version.

It is noticed that the novelty of our paper is questioned by Reviewer **HBDj** and **phCA**, here is our response for this issue:

* The novelty of our paper lies in the theoretical analysis and the UOT approach to selection of pre-training (source) data during fine-tuning. There are admittedly existing methods which select source data for a target task and we discuss the fundamental difference between existing methods and ours in Line 89-107. More importantly, the reason why the selected pre-training data are helpful to a downstream task is unveiled by our theoretical analysis (Theorem 1), which is not studied before as far as we know. Informed by Theorem 1, we design the UOT data selection to reduce the gap between source and target domain, which is commented as novel by Reviewer HBDj. In addition, as far as we know, there are few existing methods that target the fine-tuning data selection for self-supervised pre-trained models. Overall, we believe the major contributions of our work have sufficient novelty.


We hope that our response and the updated version address your concerns and our paper can be considered as qualified to be accepted.

---

### Meta-Review · Area_Chair_nB64 · 2022-08-27

**Recommendation:** Accept
**Confidence:** Certain

**Metareview:**

The paper studies reuse of source data (originally used for pre-training) in the fine-tuning phase. Due to the difference between source and target data, use of the entire source data for fine-tuning can degrade generalization for the target task. However, the paper shows that by carefully choosing a subset of the source data, the generalization performance can exceed what fine-tuning on target data alone can achieve. The scheme used for subset selection is based on unbalanced optimal transport and is theoretically justified via a Theorem in the paper. Empirical results on different datasets show that the proposed scheme indeed adds some gain in generalization.
The authors and reviewers were engaged in active discussion. Reviewers raised interesting questions including, when the source data and really benefit learning the target task, choice of neural architectures, relations to catastrophic forgetting, sensitivity to hyperparameters, usefulness of Euclidean distance for clustering in high dimensions, and practicality of the assumption that both pre-trained model, and its data are available at the fine-tuning time.
Authors provided a thorough answer to these questions. Reviewer 51Vh who was the most skeptical raised their score after the rebuttal. While the paper's final score ends up being in borderline, all the scores are on the accept side. I think the contributions of the paper are interesting enough to be published, and i recommend accept. I encourage authors to incorporate the feedback they received from the reviewers in the final version of the paper.

**Award:**

No

---

### Decision · Program_Chairs · 2022-09-14

Accept